# Full Regression of Peyronie’s Disease Plaque Following Combined Antioxidant Treatment: A Three-Case Report

**DOI:** 10.3390/antiox11091661

**Published:** 2022-08-26

**Authors:** Gianni Paulis, Giovanni De Giorgio

**Affiliations:** 1Peyronie’s Care Center, Department of Uro-Andrology, Castelfidardo Clinical Analysis Center, 00185 Rome, Italy; 2Ultrasound Diagnostics Section, Department of Uro-Andrology, Castelfidardo Clinical Analysis Center, 00185 Rome, Italy

**Keywords:** Peyronie’s disease, antioxidants, oxidative stress

## Abstract

Peyronie’s disease (PD) is a fibrotic disorder of the tunica albuginea of the penis. Conservative medical therapy includes oral and/or injective active substances. Until now, only two PD patients who recovered after medical treatment are described in the literature. This article describes three new cases of PD patients who achieved complete resorption of plaque following antioxidant treatment. Case Presentations: *Case 1.* The patient was a 34-year-old man, a smoker, with lateral-left penile curvature (40 degrees), penile pain, and good penile rigidity. The patient was treated with combined therapy (oral antioxidants + Propolis creme). At follow-up, after about 29 months of treatment, we observed the disappearance of the acquired penile deformity. The ultrasound examination no longer showed any plaque. *Case 2.* The patient was a 32-year-old man with chronic prostatitis, penile lichen sclerosus, lateral-left penile curvature (10 degrees), a palpable lump, and good penile rigidity. The patient was treated with combined therapy (oral antioxidants + Propolis creme + penile injections of pentoxifylline). After 33 months of treatment, at follow-up, we observed the disappearance of the penile deformity. Ultrasound examinations no longer showed any plaque. *Case 3.* The patient was a 33-year-old man with penile pain, dorsal penile curvature (30 degrees), and good penile rigidity. The patient was treated with combined therapy (oral antioxidants + Propolis creme + penile injections of pentoxifylline). At follow-up, after 41 months of treatment, the penile pain was no longer present, and the ultrasound study no longer showed any plaque. Conclusions: Although our study presents a limited number of cases, it is a matter of fact that these patients obtained full regression in the affected area. We believe our experience may be very useful for urological clinical practice.

## 1. Introduction

Peyronie’s disease (PD) affects adult men and is characterized by chronic inflammation of the penis (tunica albuginea). Its hallmark is the growth of fibrous plaque, which in certain cases becomes calcified and causes penile deformation. Symptoms of PD include penile deformity (curvature, shortening, divots, hourglass, etc.), penile pain, erectile dysfunction, and anxiety–depression [1,2,3]. The prevalence of PD varies between 3.2 and 13% [4,5]. Although the etiopathogenesis of the disease has not been fully explained yet, it is known that PD affects males with a genetic predisposition, and the triggering factor appears to be trauma, which need not necessarily be severe, and it is more or less associated with the presence of risk factors (diabetes, high blood pressure, chronic prostatitis, autoimmune disorders, etc.) [6,7,8,9]. Following the traumatic event, an accumulation of blood and fibrin forms in situ, causing the onset of the disease, with a strong recruitment of inflammatory cells leading to the overproduction of fibrogenic factors and reactive oxygen species (oxidative stress) [10,11]. It has recently been proved that the latter, in particular, plays a decisive role in the formation of plaque and the evolution of the disease itself [11,12,13]. Whereas surgery is the treatment of choice for stable-phase disease, in situations involving severe erectile dysfunction and/or severe penile curvature, which prevent penetration, conservative treatment is indicated in the active phase of the disease (inflammatory stage) and includes the following: oral vitamin E, potaba, colchicine, tamoxifen, antioxidants, and phosphodiesterase-5 inhibitors; penile-injections of anti-inflammatory agents, anti-fibrotic substances, and antioxidants (verapamil, corticosteroids, pentoxifylline, collagenase, interferon, etc.); and physical therapy (iontophoresis, ESWT, vacuum devices, and penile traction devices) [14,15,16]. This report presents three cases of patients with PD who were followed at our “PD Care Center” and who achieved full plaque resolution following multimodal treatment with antioxidants. This case report follows another article in which we described two other cases of full recovery from the disease following antioxidant treatment.

## 2. Case Presentations

### 2.1. Case 1

A 34-year-old Caucasian man, a smoker (10 cigarettes per day), with congenital lateral-left penile curvature (10 degrees before the onset of PD), penile pain, and normal penile rigidity presented at our clinic in April 2019, complaining of pain in the penis during erections and curvature of the penis which had begun 6 months prior. The visual analogue pain score (VAS) was 2 (scored from 0 to 10). The score on the International Index of Erectile Function (IIEF) questionnaire was 26. When administering the IIEF questionnaire, we took into consideration questions 1, 2, 3, 4, 5, and 15, which refer to the aspect of *erectile function* (normal range from 26 to 30). A penile Doppler ultrasound was performed (alprostadil 10 mcg). The penile deformity consisted of a lateral-left penile curvature of 40 degrees.

The arterial flow of the cavernous arteries and the end-diastolic velocity were normal: peak-systolic velocity = 92 cm/s (left) and 90 cm/s (right); end-diastolic velocity = 0 cm/s (on both sides). Three-dimensional measurements of the volume of plaque were taken using an ellipsoid formula (volume = 0.524 × width × length × thickness) [17,18].

The plaque was located in the middle of the penis, its ultrasound aspect was iso-hyperechoic, and it measured 10.8 × 10.1 × 3.03 mm (174 mm^3^ = volume) (Figure 1, Table 1).

During the informed consent process, the patient was told that treatment would necessarily be long due to the chronic nature of the disease. The patient refused to the publication of photos of his penis, even if they were anonymous. The patient refused to be treated with penile infiltrations for fear of pain. Beginning in April 2019, after giving his informed consent, the patient was treated with combined therapy: oral antioxidants silymarin 400 mg + Ginkgo biloba 250 mg + propolis 600 mg + bilberry 160 mg + vitamin E 800 IU/once a day/+ propolis creme/twice a day/for 12 months.

At the end of the first treatment cycle, at follow-up, the patient filled out the IIEF questionnaire, and the score was 26. We then observed a left penile curvature with a decreased angle (to 25 degrees). The pain in the penis had gone. The penile ultrasound showed the following dimensions: 6.33 × 5.58 × 2.66 mm (49 mm^3^ = volume) (Figure 2).

Compared to its initial measurement, the plaque had therefore undergone a 71.8% reduction in volume. Considering the excellent result after the first therapy cycle, we repeated the treatment for another 6 months.

After the second treatment cycle, the IIEF score at follow-up was unchanged at 26, and we observed that the lateral-left penile curvature had an angle reduced to 20 degrees. The ultrasound showed the following dimensions: 3.98 × 3.33 × 2.20 mm (15 mm^3^ = volume) (Figure 3).

The volume of the plaque was thus 91.3% smaller. Considering the excellent result after the second therapy cycle, we decided to continue the same multimodal treatment for only another 6 months. However, the patient continued the same treatment for 9 months.

After three treatment cycles, for an overall 2 years and 5 months (29 months) of multimodal treatment, the IIEF score of the patient at follow-up was the same (scoring 26). We observed that the lateral-left penile curvature had an angle reduced to 10 degrees, similar to the curve present before PD (congenital curve). The ultrasound examination no longer showed any plaque (Figure 4). 

The ultrasonography was performed by the same physician at the patient’s initial presentation, as well as at the later follow-ups, and always using the same device (Philips Affinity 70 G).

The multimodal therapy with antioxidants was suspended. No side effects were reported by the patient after treatment.

### 2.2. Case 2

A 32-year-old Caucasian man, a non-smoker, with chronic prostatitis, penile lichen sclerosus, lateral-left penile curvature (10 degrees), penile pain, and normal penile rigidity, presented at our clinic in March 2017. At the time of our observation, the visual analogue pain score (VAS) was 3 (scored from 0 to 10).

The IIEF score was 26. Physical examination revealed, on palpation of the proximal third of the penis, a nodule measuring about 20 mm in length. A penile Doppler ultrasound was performed (alprostadil 10 mcg). The penile deformity consisted of lateral-left penile curvature (10 degrees). The arterial flow of the cavernous arteries and the end-diastolic velocity were normal: peak-systolic velocity = 95 cm/s (left) and 92 cm/s (right); end-diastolic velocity = 0 cm/s (on both sides). The plaque was located at the base of the penis, its ultrasound aspect was iso-hyperechoic, and it measured 16.4 × 8.27 × 3.09 mm (219 mm^3^ = volume) (Table 1, Figure 5).

During the informed consent process, the patient was told that treatment would necessarily be long due to the chronic nature of the disease. The patient refused to the publication of photos of his penis, even if they were anonymous.

Beginning in May 2017, after giving his informed consent, the patient was treated with combined therapy: oral antioxidants silymarin 400 mg + Ginkgo biloba 250 mg + propolis 600 mg + bilberry 160 mg + vitamin E 800 IU/once a day/+ propolis creme/twice a day/+ penile injection (peri-plaque) with pentoxifylline (100 mg, using a 30G needle) every 15 days for 6 months.

At the end of the first treatment cycle, at a follow-up, the patient filled out the IIEF questionnaire, and the score was observed to have increased to 28.

We then observed a left penile curvature with a decreased angle (to 6 degrees). The penile pain was gone. In the previous examination, we used the ultrasound machine Philips HD 15; however, subsequently, the ultrasound machine in our clinic was replaced by a new machine (Philips Affinity 70 G), which was used in the latter and in subsequent follow-ups. The same doctor performed all the ultrasound examinations. The penile ultrasound showed the following plaque dimensions: 12.9 × 6.66 × 3.01 mm (135 mm^3^ = volume) (see Figure 6).

Compared to its initial measurement, the plaque had therefore undergone a 38.3% reduction in volume.

Considering the good result after the first therapy cycle, we repeated the same treatment for another 12 months (second cycle), while reducing the pentoxifylline 100 mg injection frequency to one penile peri-plaque injection every month.

At the end of the second cycle of treatment, the patient underwent follow-up with a physical exam and a penile Doppler ultrasound. At follow-up, the penile pain was no longer present, and the IIEF score was 28. We then observed a lateral-left penile curvature with an unchanged angle (6 degrees).

The penile ultrasound showed the following plaque dimensions: 9.27 × 5.86 × 2.58 mm (73 mm^3^ = volume) (Figure 7).

Compared to its initial measurement, the plaque had therefore undergone a 66.6% reduction in volume.

In consideration of the good response achieved, even with the second cycle of multimodal therapy, the decision was made to continue the same oral and topical treatment for a third cycle, while reducing the pentoxifylline 100 mg injection frequency to one penile peri-plaque injection every other month.

After the completion of the third treatment cycle, and after 2 years and 9 months (33 months), the patient underwent follow-up with a physical exam and a penile Doppler ultrasound. At follow-up, the IIEF score was 28. The penile pain was no longer present. We then observed an absence of penile deformation. The ultrasound examination no longer showed any plaque (Figure 8). 

The multimodal therapy with antioxidants was suspended. No side effects were reported by the patient after treatment. The same physician performed all the ultrasonography examinations.

### 2.3. Case 3

A 33-year-old Caucasian man, a non-smoker, with dorsal penile curvature (30 degrees), a palpable nodule, and normal penile rigidity, presented at our clinic in May 2017, complaining of penile pain during erection, with an onset of about 1 year before.

The visual analogue pain score (VAS) was 2. The International Index of Erectile Function score was 26. On penile palpation, there was no evidence of any penile nodules.

A penile Doppler ultrasound was performed (alprostadil 10 mcg). The penile deformity consisted of dorsal penile curvature (30 degrees). The arterial flow of the cavernous arteries and the end-diastolic velocity were normal: peak-systolic velocity = 96 cm/s (left) and 94 cm/s (right); end-diastolic velocity = 0 cm/s (on both sides). The plaque was located in the middle of the penis, its ultrasound aspect was iso-hypoechoic, and it measured 20.6 × 15.2 × 4.42 mm (724 mm^3^ = volume); within the plaque, there were some small calcifications, the largest of which measured 1.9 × 4.1 mm (Figure 9). 

During the informed consent process, the patient was told that treatment would necessarily be long due to the chronic nature of the disease. The patient refused to the publication of photos of his penis, even if they were anonymous.

Beginning in May 2017, after giving his informed consent, the patient was treated with combined therapy: oral antioxidants silymarin 400 mg + Ginkgo biloba 250 mg + propolis 600 mg + bilberry 160 mg + vitamin E 800 IU/once a day/+ propolis creme/twice a day/+ penile injection (peri-plaque) of pentoxifylline (100 mg, using a 30 G needle) every 15 days for 6 months.

After the first treatment cycle, at follow-up, the patient filled out the IIEF questionnaire, and the score was 26. We then observed a dorsal penile curvature with a decreased angle (to 25 degrees). The penile pain was gone.

In the previous examination, we used the ultrasound machine Philips HD 15; subsequently, the ultrasound machine in our clinic was replaced by a new machine (Philips Affinity 70 G), which was used in the latter and subsequent follow-ups. The same doctor performed all the ultrasound examinations. The penile ultrasound showed the following dimensions: 14.1 × 8.54 × 3.74 mm (236 mm^3^ = volume) and the disappearance of the internal calcifications (Figure 10).

Compared to its initial measurement, the plaque had therefore undergone a 67.4% reduction in volume.

Considering the good result after the first therapy cycle, we repeated the same treatment for another 12 months (second cycle), while reducing the pentoxifylline 100 mg injection frequency to one penile peri-plaque injection every month.

At the end of the second cycle of treatment, the patient underwent follow-up with a physical exam and a penile Doppler ultrasound. At follow-up, the IIEF score was 27. We then observed a dorsal penile curvature with an angle reduced to 15 degrees. The penile ultrasound showed the following plaque dimensions: 9.14 × 7.62 × 1.27 mm (127 mm^3^ = volume) (Figure 11).

Compared to its initial measurement, the plaque had therefore undergone an 82.4% reduction in volume. In consideration of the good response achieved, even with the second cycle of multimodal therapy, the decision was made to continue the same oral and topical treatment for a third cycle, while reducing the pentoxifylline 100 mg injection frequency to one penile peri-plaque injection every other month.

After the third treatment cycle, the patient underwent a similar follow-up with a physical exam and a penile Doppler ultrasound. The IIEF score was 27. We then observed a dorsal penile curvature of 15 degrees (unchanged). The penile ultrasound showed the following plaque dimensions: 4.95 × 3.91 × 2.02 mm (21 mm^3^ = volume) (Figure 12).

Compared to its initial measurement, the plaque had therefore undergone a 97% reduction in volume.

Considering the good result after the third therapy cycle, we repeated the same treatment (oral antioxidants + creme) for another 6 months (fourth cycle) and suspended the pentoxifylline injections.

After the fourth treatment cycle, the patient underwent a similar follow-up, and after 3 years and 5 months of multimodal treatment, the patient underwent the same follow-up with a physical exam and a penile Doppler ultrasound. The IIEF score was 27. We then observed an absence of penile deformation. The ultrasound examination no longer showed any plaque (Figure 13).

The multimodal therapy with antioxidants was suspended. No side effects were reported by the patient after treatment. The same physician performed all ultrasonography examinations.

## 3. Discussion

In the literature, a number of studies have already described a reduction in PD plaque and an improvement in penile deformity [19,20,21,22]. In a recent article, we described two cases of PD patients who achieved a complete disappearance of plaque following multimodal treatment with antioxidants [23]. Before the publication of our last article, plaque disappearance in PD had already been reported, but in an article describing an experimental study on rats and not on men [24]. It may be noticed that our treatment required a lengthy period of time; this was necessary because PD is a “chronic” inflammatory disease and, consequently, complete resorption of the diseased area (plaque) requires a length of time that depends on the size of the plaque.

Although our article describes a limited number of cases, we think it is of great importance for the practice of all urologists.

We believe the success of our treatment lies in the properties of the antioxidant substances we used; these substances can interrupt the inflammatory process, combating oxidative stress, which has been shown to play an essential role in the production of fibrogenic factors [10]. All the antioxidants used in our combined treatment also have anti-inflammatory and anti-fibrotic properties, which block the production of pro-inflammatory cytokines, thanks to their inhibiting activity toward factor NF-kB [25,26,27,28,29].

Propolis, bilberry, silymarin, and Ginkgo biloba have the following action properties: antioxidant and anti-fibrotic activity; the inhibition of proinflammatory cytokines IL-6, IL-8, TGF-beta 1, IL-1, and TNF-α; the inhibition of MMP-2 and MMP-9 (with anti-elastatic activity); the inhibition of the COX-2 enzyme; the inhibition of the activation of the NF-kappa-B factor; PDGF-activity inhibition, etc. [12,26,27,28,29].

Vitamin E has the following action properties: scavenger activity against ROS, hydroxyl radicals, and lipid peroxyl radicals; NFK-B factor inhibition; proinflammatory cytokine-expression inhibition; COX-2 inhibition; PDGF-activity inhibition; and the inhibition of fibroblast proliferation [12].

Pentoxifylline has the following action properties: the inhibition of ROS and lipid oxidation; the inhibition of myofibroblastic differentiation of fibroblasts; a reduction in collagen deposition; NFK-B factor inhibition; proinflammatory cytokine-expression inhibition; COX-2 inhibition; TGF-beta-1 inhibition; PAI 1-expression inhibition; fibroblast apoptosis stimulation; non-specific PDE inhibitory power; and the downregulation of iNOS protein expression [12,25].

It may have been noticed that our treatment cycles call for a gradual lengthening of the interval between pentoxifylline injections. This treatment plan is dictated by the fact that even a simple perilesional injection represents a trauma, however minimal, and trauma is unquestionably the factor that triggers the disease [7,8]. Therefore, any time that follow-up showed a partial regression of PD, we found it right to reduce the possibility of any new trauma, however minimal, by more widely spacing out the injections.

Our multimodal therapy allowed penile plaque to regress and penile curvature improvement. In our treatment, the anti-inflammatory activity was completely natural and without the use of classic anti-inflammatories (non-steroidal anti-inflammatory/NSAIDs and steroidal substances). In the majority of cases, classic anti-inflammatory substances fail to cure PD; this is possibly due to a delayed onset of treatment, failure to adequately diagnose the presence of plaque, or inadequate treatment compliance.

With respect to diagnostic imaging, although sonographic assessment of the penis has been reported in the literature to be unable to provide adequate plaque measurements, we believe, instead, that accurate plaque size can be obtained if a highly sensitive, state-of-the-art ultrasound machine is employed, and, above all, if the evaluation is performed by a physician experienced in the diagnosis of this disease [30,31,32]. We, therefore, believe we owe our results not only to the substances used in the course of the therapy, but also to the characteristics of our ultrasound assessment, which enabled us to accurately diagnose the area affected by the disease (plaque) and constantly monitor the evolution of the disease at scheduled follow-ups.

## 4. Conclusions

Although our study presents a very limited number of cases, it is a matter of fact that these patients obtained full regression in the affected area.

We believe our success was due essentially to three important factors: the type of antioxidant substances; the use of an extremely sensitive ultrasound machine, which was able to provide the precise dimensions of the plaque; and, moreover, the fact that the scan was performed by a physician with extensive experience in Peyronie’s disease. We believe our experience, however limited, may be very useful for the clinical practice of specialists in the field of uro-andrology. In any case, randomized controlled studies with a greater number of cases are desirable and needed to prove the effectiveness of this type of treatment for PD.

## Figures and Tables

**Figure 1 antioxidants-11-01661-f001:**
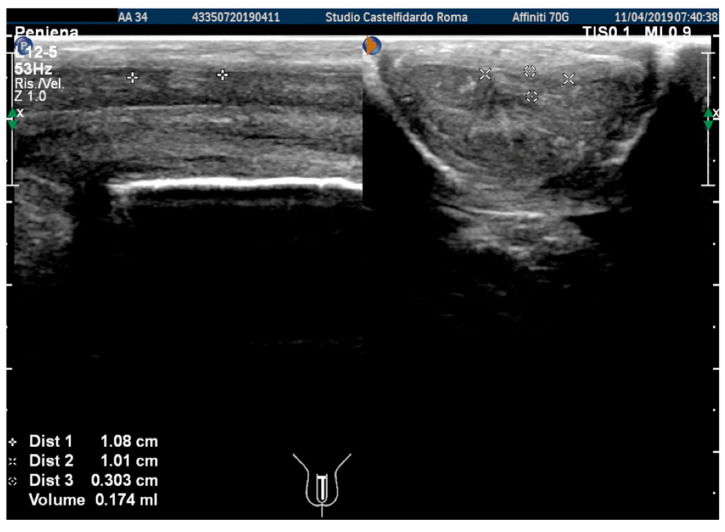
Ultrasonography of the penis before therapy (longitudinal and axial views).

**Figure 2 antioxidants-11-01661-f002:**
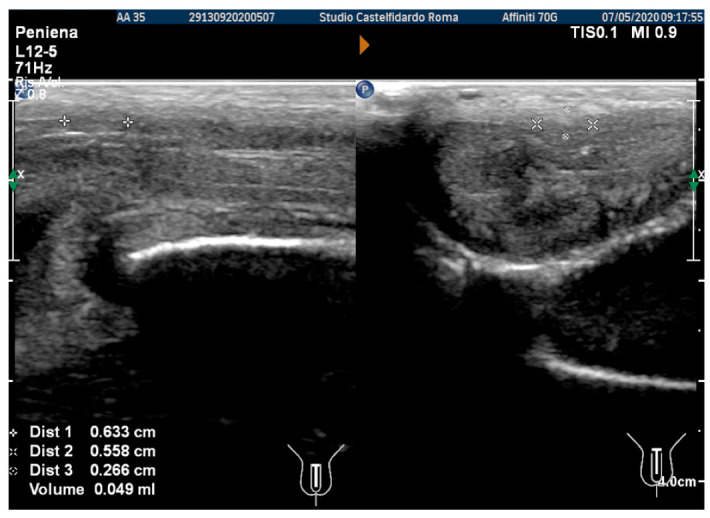
Ultrasonography of the penis after the 1st therapy cycle (longitudinal and axial views).

**Figure 3 antioxidants-11-01661-f003:**
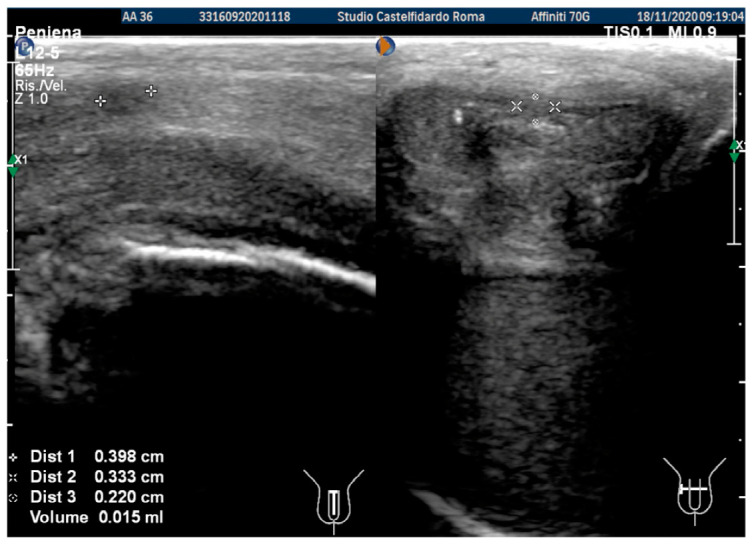
Ultrasonography of the penis after the 2nd therapy cycle (longitudinal and axial views).

**Figure 4 antioxidants-11-01661-f004:**
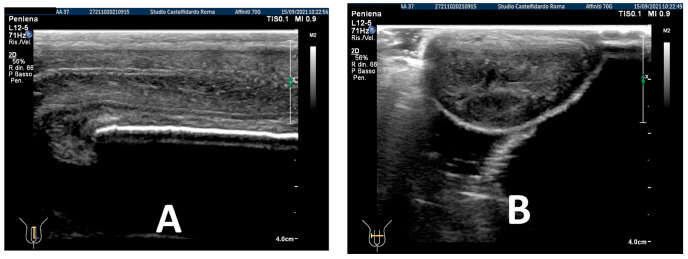
Ultrasonography of the penis after the 3rd therapy cycle. (**A**) Longitudinal view; (**B**) axial view.

**Figure 5 antioxidants-11-01661-f005:**
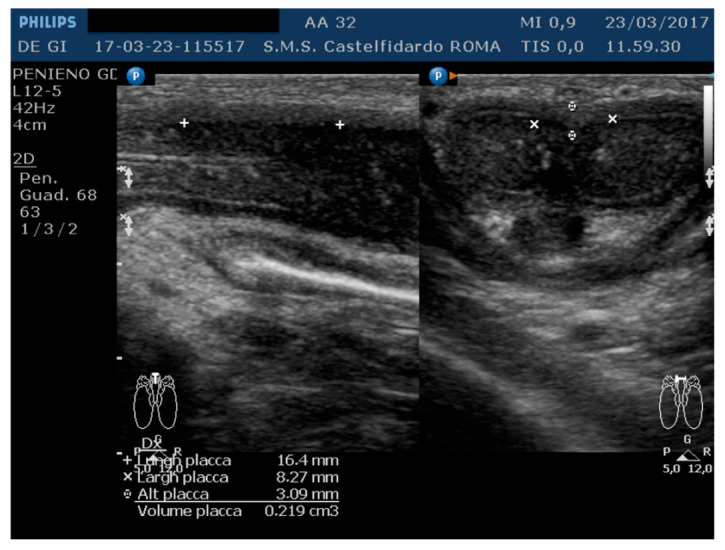
Ultrasonography of the penis before therapy (longitudinal and axial views).

**Figure 6 antioxidants-11-01661-f006:**
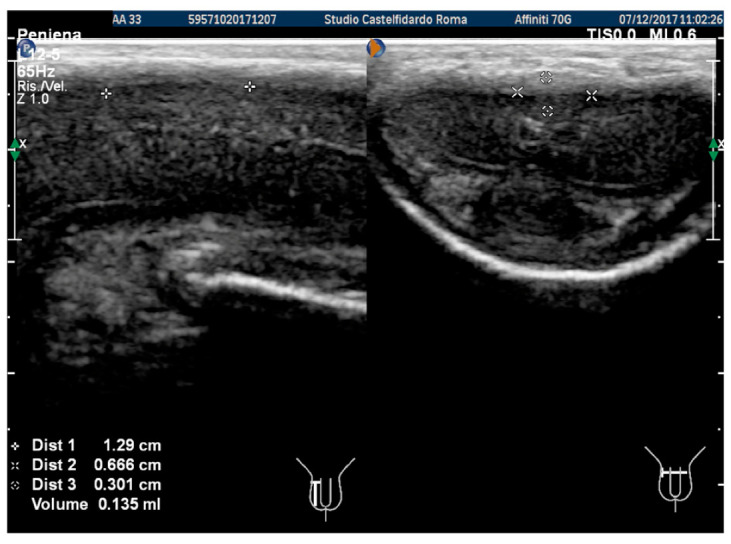
Ultrasonography of the penis after the 1st therapy cycle (longitudinal and axial views).

**Figure 7 antioxidants-11-01661-f007:**
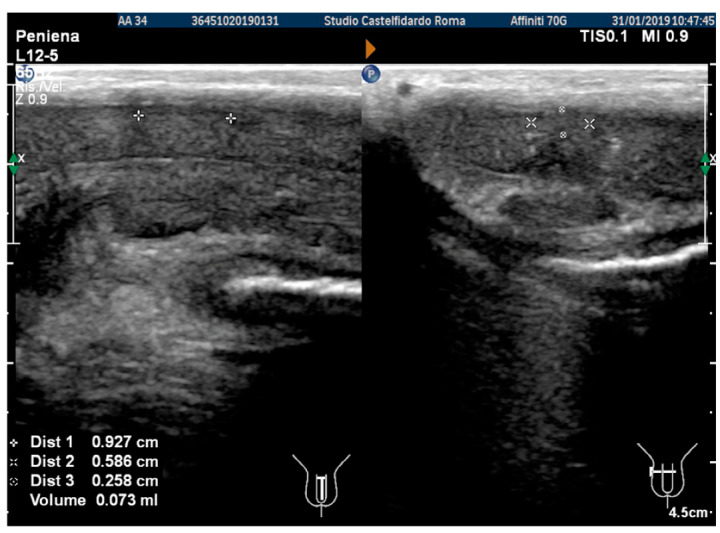
Ultrasonography of the penis after the 2nd therapy cycle (longitudinal and axial views).

**Figure 8 antioxidants-11-01661-f008:**
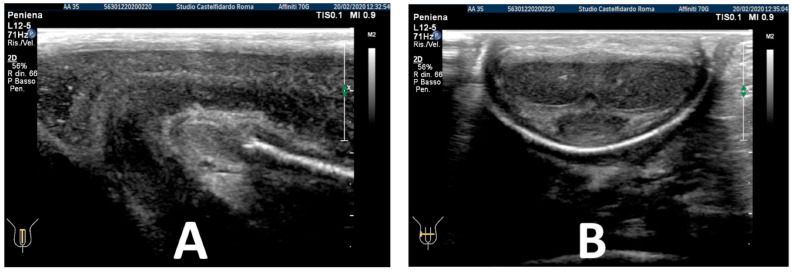
Ultrasonography of the penis after the 3rd therapy cycle. (**A**) Longitudinal view; (**B**) axial view.

**Figure 9 antioxidants-11-01661-f009:**
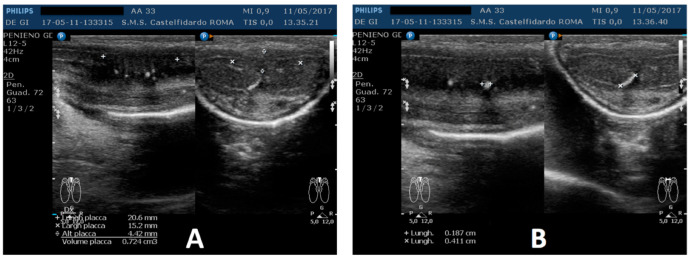
Ultrasonography of the penis before therapy (longitudinal and axial views). (**A**) Entire plaque; (**B**) the largest calcification.

**Figure 10 antioxidants-11-01661-f010:**
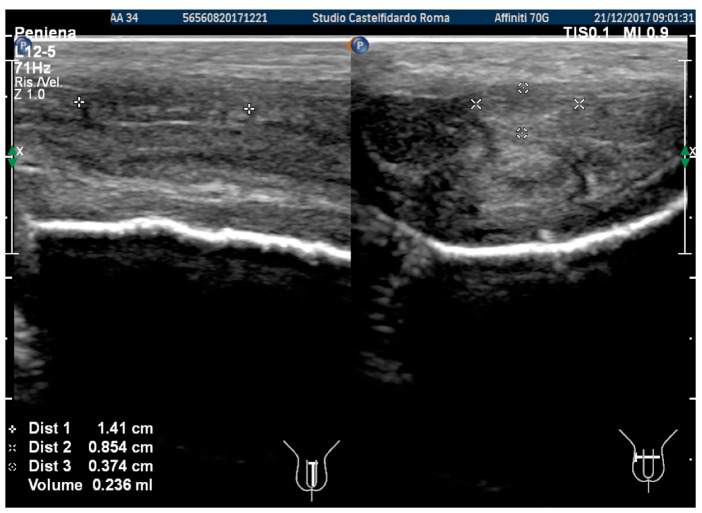
Ultrasonography of the penis after the 1st therapy cycle (longitudinal and axial views).

**Figure 11 antioxidants-11-01661-f011:**
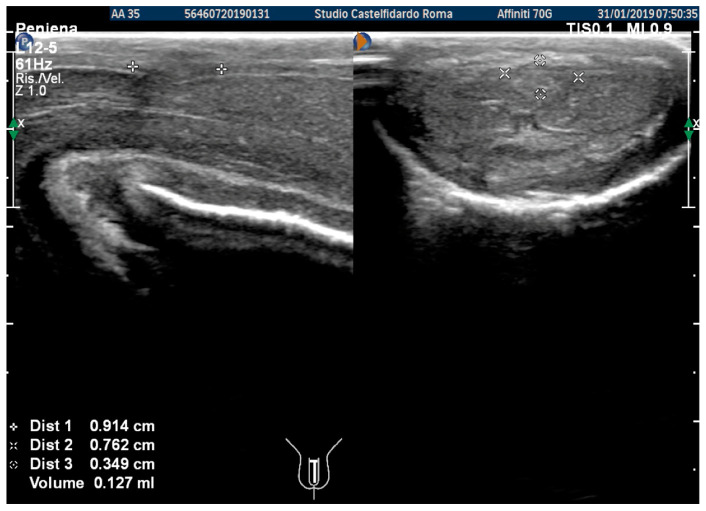
Ultrasonography of the penis after the 2nd therapy cycle (longitudinal and axial views).

**Figure 12 antioxidants-11-01661-f012:**
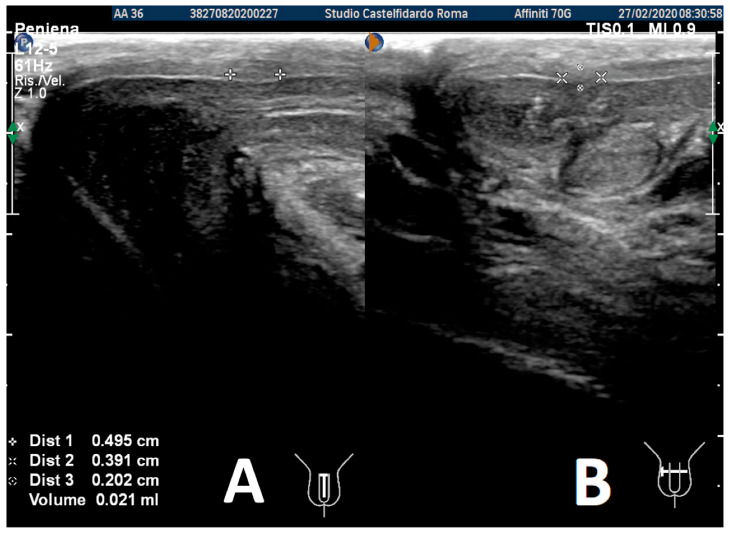
Ultrasonography of the penis after the 3rd therapy cycle. (**A**) Longitudinal view; (**B**) axial view.

**Figure 13 antioxidants-11-01661-f013:**
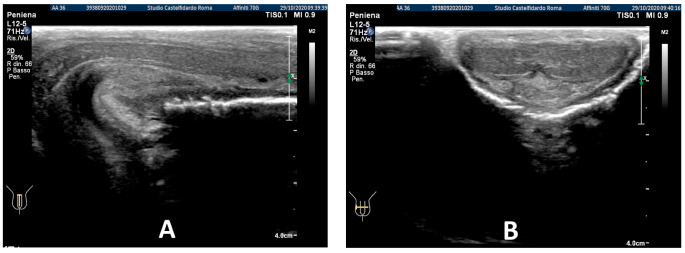
Ultrasonography of the penis after the 3rd therapy cycle. (**A**) Longitudinal view; (**B**) axial view.

**Table 1 antioxidants-11-01661-t001:** Summary of the 3 cases of patients with Peyronie’s disease who were treated with multimodal antioxidant therapy.

#	Age/Years	Concomitant Diseases	Localization of the Penile Plaque	Plaque Measurements ( Length × Width × Thickness) and Volume of the Plaque (A) Prior and (B) After Therapy	Typology of the Curvature (A) Prior and (B) After Therapy	Penile Pain VAS Scale (Score 1 to 10) (A) Prior and (B) After Therapy	IIEF Questionnaire Score (A) Prior and (B) After Therapy	Duration of Therapy Up to Plaque Regression	Description of the Administered Multimodal Therapy
1	34 years	Congenital lateral-left penile curvature (10-degrees)	Middle third	(A) 10.8 × 10.1 × 3.03 mm volume = 174 mm^3^	(A) 40-degree left curvature	(A) score 2	(A) score 26	2 years and 5 months	orally: Silymarin 400 mg + Ginkgo biloba 250 mg + Propolis 600 mg + Bilberry 160 mg + Vitamin E 800 IU/once a day, for 29 months. + topically: Propolis creme/twice a day/for 29 months. * The patient refused peri-plaque penile injections
				(B) No plaque detected	(B) 10-degree left penile curvature (previous condition = congenital lateral left penile curvature)	(B) score 0 (after 12 months)	(B) score 26		
2	32 years	lichen sclerosus, chronic prostatitis	Proximal third	(A) 16.4 × 8.27 × 3.09 mm volume = 219 mm^3^	(A) 10-degree left curvature	(A) score 3	(A) score 26	2 years and 9 months	orally: Silymarin 400 mg + Ginkgo biloba 250 mg + Propolis 600 mg + Bilberry 160 mg + Vitamin E 800 IU/once a day, for 33 months. + topically: Propolis creme/twice a day/for 33 months. + peri-plaque penile injections: Pentoxifylline 100 mg (30 G needle) every 15 days for 6 months, and then monthly for 12 months, and then 1 injection every other month. for 12 months (total = 30 injections)
				(B) No plaque detected	(B) None	(B) score 0 (after six months)	(B) score 28		
3	33 years	None	Middle third	(A) 20.6 × 15.2 × 4.42 mm volume = 724 mm^3^ + some small calcifications, the largest of which measured 1.9 × 4.1 mm	(A) 30-degree dorsal penile curvature	(A) score 2	(A) score 26	3 years and 5 months	orally: Silymarin 400 mg + Ginkgo biloba 250 mg + Propolis 600 mg + Bilberry 160 mg + Vitamin E 800 IU/once a day, for 41 months. + topically: Propolis creme/twice a day/for 41 months. + peri-plaque penile injections: Pentoxifylline 100 mg (30 G needle) every 15 days for 6 months, and then monthly for 12 months, and then 1 injection every other month. for 12 months (total = 30 injections)
				(B) No plaque detected	(B) None	(B) score 0 (after six months)	(B) score 27		

* Abbreviations: IIEF score = Erectile function scoring; VAS = Pain scoring.

## Data Availability

Not applicable.

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
