# Peer review of "Full Regression of Peyronie’s Disease Plaque Following Combined Antioxidant Treatment: A Three-Case Report"

_antioxidants, 2022, doi:10.3390/antiox11091661_

Round 1

Reviewer 1 Report

I found this paper interesting, well written, although the study is limited by the number of cases that do not allow us to draw firm conclusion about the effect of the drugs used. In particular you didn't use the same protocol for the three patients and the use of Vitamine E and pentoxifylline is not raccomanded in the EAU Guidelines because of lack of evidence. I think it would be interesting if you extend your study to more patients to draw conclusion about a new protocol and to prove the effectiveness of this type of treatment for PD.

Reviewer 2 Report

1. The Corresponding Author's email address is missing an @.

2. Line 13: maybe a "who" is missing? "PD-patients WHO recovered"...

3. Line 22: either remove the parentheses before pentoxifylline, or add a second one after.

4. Line 26: as #3

5. Line 29, remove the comma (our experience may be...).

Introduction: providing some additional information on the prevalence of PD in the general population and among patients attending andrological facilities might be useful to the reader.

Discussion: I would suggest some minor improvements in this section.
First, it would be beneficial for the overall quality of the paper to disclose more in detail the possible mechanisms of action of the different compounds used in the present case series. Indeed, as you correctly pointed out, the use of different anti-inflammatory molecules can have an effect on plaque resorption, and understanding the different pathways involved could be a key factor in order to provide more adequate (and tailored) treatment.
Second, I would suggest pointing out that while these treatments greatly improved the penile curvature of the three patients herein described, in most cases anti-inflammatory treatments do not yield such results. This is possibly due to delayed onset of treatment, failure to adequately diagnose the presence of a plaque, or inadequate treatment compliance.

Overall the manuscript is well-written and provides useful insight into a common, yet largely underdiagnosed and undertreated condition.
